

# Reproducible and reusable research: are journal data sharing policies meeting the mark?

Nicole A. Vasilevsky[1,2], Jessica Minnier[3], Melissa A. Haendel[1,2] and Robin E. Champieux[1]

[1] OHSU Library, Oregon Health & Science University, Portland, OR, United States
[2] Department of Medical Informatics and Clinical Epidemiology, Oregon Health & Science University, Portland, OR, United States
[3] OHSU-PSU School of Public Health, Oregon Health & Science University, Portland, OR, United States

## ABSTRACT

**Background**. There is wide agreement in the biomedical research community that research data sharing is a primary ingredient for ensuring that science is more transparent and reproducible. Publishers could play an important role in facilitating and enforcing data sharing; however, many journals have not yet implemented data sharing policies and the requirements vary widely across journals. This study set out to analyze the pervasiveness and quality of data sharing policies in the biomedical literature.

**Methods**. The online author's instructions and editorial policies for 318 biomedical journals were manually reviewed to analyze the journal's data sharing requirements and characteristics. The data sharing policies were ranked using a rubric to determine if data sharing was required, recommended, required only for omics data, or not addressed at all. The data sharing method and licensing recommendations were examined, as well any mention of reproducibility or similar concepts. The data was analyzed for patterns relating to publishing volume, Journal Impact Factor, and the publishing model (open access or subscription) of each journal.

**Results**. A total of 11.9% of journals analyzed explicitly stated that data sharing was required as a condition of publication. A total of 9.1% of journals required data sharing, but did not state that it would affect publication decisions. 23.3% of journals had a statement encouraging authors to share their data but did not require it. A total of 9.1% of journals mentioned data sharing indirectly, and only 14.8% addressed protein, proteomic, and/or genomic data sharing. There was no mention of data sharing in 31.8% of journals. Impact factors were significantly higher for journals with the strongest data sharing policies compared to all other data sharing criteria. Open access journals were not more likely to require data sharing than subscription journals.

**Discussion**. Our study confirmed earlier investigations which observed that only a minority of biomedical journals require data sharing, and a significant association between higher Impact Factors and journals with a data sharing requirement. Moreover, while 65.7% of the journals in our study that required data sharing addressed the concept of reproducibility, as with earlier investigations, we found that most data sharing policies did not provide specific guidance on the practices that ensure data is maximally available and reusable.

Corresponding author
Robin E. Champieux,
champieu@ohsu.edu

## INTRODUCTION

Over the last several years, the importance and benefits of research data sharing have been emphasized by many communities, including professional societies, funders, policy makers, and publishers (*NIH, 2016b*; *Holdren, 2012*; *Research Councils UK, 2011*; *Drazen et al., 2016*; *Medium.com, 2016*). Several rationales underpin the arguments for better access to and the curation of research data (*Borgman, 2012*). While the factors contributing to the poor reproducibility of biomedical research are varied and complex, and even the meaning of reproducible research is fraught, data availability is regarded as one necessary component for the assessment of replication and validation studies (*Collins & Tabak, 2014*). If raw data are made available, others have the opportunity to replicate or correct earlier findings and, ostensibly, influence the pace and efficiency of future research endeavors. Researchers can ask new questions of existing data, and data can be combined and curated in ways that further its value and scholarship (*Borgman, 2012*). As Fischer and Zigmond argue, the great advances in science depend not only on the contributions of many individual researchers, but also their willingness to share the products of their work (*Fischer & Zigmond, 2010*).

The benefits described above have motivated many of the organizations that support research to require that data be made publicly available. Since 2011, the National Science Foundation (NSF) has required applicants to submit a data management plan documenting how investigators will conform to the NSF's expectation that primary data and research resources will be shared with other researchers (*NSF, 2016*). The White House Office of Science and Technology Policy issued a memorandum in 2013 directing agencies to make plans for ensuring public access to federally funded research results, including data (*Holdren, 2012*). In 2014, the National Institutes of Health (NIH) implemented a strong data sharing policy for large-scale human and non-human genomic data (*NIH, 2016a*). Additionally, the European Research Council's Open Access Guidelines include and support public access to research data, and open is the default for all data generated via its Horizon 2020 program (*European Commission, 2016*).

However, data sharing and its long-term stewardship involve an array of activities, participants, and technologies, especially if discovery, reuse, and preservation are to be ensured (*Dallmeier-Tiessen et al., 2014*). Moreover, despite a belief in the importance of access to other's data for their own work, many scientists do not consistently share their data, reporting a variety of barriers and disincentives (*Tenopir et al., 2011*). Roadblocks to sharing include insufficient time, a lack of funding, fear of scrutiny or misinterpretation, a deficit of requirements, attribution concerns, competition, difficulty navigating infrastructure options, and a paucity of data sharing related rewards (*Longo & Drazen, 2016*; *LeClere, 2010*; *Savage & Vickers, 2009*). For quality data sharing to become the norm, broad systemic change and solutions are needed.

Journal publication is the current and primary mode of sharing scientific research. While arguably problematic, it has the most influence on an individual's credibility and success (*Fischer & Zigmond, 2010*). As Lin and Strasser write, journals and publishers occupy an important "leverage point in the research process," and are key to affecting the changes needed to realize data sharing as a "fundamental practice" of scholarly communication (*Lin & Strasser, 2014*). There has been significant support for and progress toward this end. At a joint workshop held at the NIH in June 2014, editors from 30 basic and preclinical science journals met to discuss how to enhance reproducible, robust, and transparent science. As an outcome, they produced the "Principles and Guidelines for Reporting Preclinical Research," which included the recommendation that journals require that all of the data supporting a paper's conclusion be made available as part of the review process and upon publication, that datasets be deposited to public repositories, and that datasets be bi-directionally linked to published articles in a way that ensures attribution (*NIH, 2016b*). In 2013, Nature journals implemented an 18-point reporting checklist for life science articles. It included required data and code availability statements, and a strong recommendation for data sharing via public repositories (*Nature Publishing Group, 2013*). Additionally, many large and influential journals and publishers have implemented data sharing requirements, including Science, Nature, the Public Library of Science (PLOS), and the Royal Society (*AAAS S, 2016*; *Nature, 2016*; *PLOS, 2016*; *The Royal Society, 2016*).

Given these developments, and the influence of journal publishing on scientific communication and researcher success, we sought to investigate the prevalence and characteristics of journal data sharing policies within the biomedical research literature. The study was designed to determine the pervasiveness and quality of data sharing policies as reflected in editorial policies and the instructions to authors. In other words, we aimed to assess the specific characteristics of the policies as they relate to data sharing in practice. We focused our analysis on the biomedical literature because of the intense attention data availability and its relationship to issues of reproducibility and discovery have received, and on account of our own roles as and work with biomedical researchers.

## MATERIALS & METHODS

We evaluated the data sharing policies of journals that were included in Thomson Reuter's InCites 2013 Journal Citations Reports (JCR) (*Clarivate Analytics, 2016*) classified within the following Web of Science schema categories: Biochemistry and Molecular Biology, Biology, Cell Biology, Crystallography, Developmental Biology, Biomedical Engineering, Immunology, Medical Informatics, Microbiology, Microscopy, Multidisciplinary Sciences, and Neurosciences. These categories were selected to capture the journals publishing the majority of peer-reviewed biomedical research. The original data pull included 1,166 journals, collectively publishing 213,449 articles. We filtered this list to the journals in the top quartiles by Impact Factor (IF) or number of articles published in 2013. Additionally, the list was manually reviewed to exclude short report and review journals, and titles determined to be outside the fields of basic medical science or clinical research. The final study sample included 318 journals, which published 130,330 articles in 2013. The study

sample represented 27% of the original Journal Citation Report list and 61% of the original citable articles. After the sample was identified, the 2014 Journal Citations Reports was released. While we did not use this data to change the journals in the study sample, we did employ data from both reports in the analyses presented here. After our analysis, the 2015 Journal Citation Reports was released. While we found no significant differences in the distribution of impact factor nor citable items by year, nor in subsets defined by data sharing marks, results for the 2015 data are available in our Github repository (https://github.com/OHSU-Library/Biomedical_Journal_Data_Sharing_Policies). In our data pulls from JCR, we included the journal title, International Standard Serial Number (ISSN), the total citable items, the total citations to the journal, Impact Factor, and the publisher. Table 1 reports the number (and percentage) of journals across 2013 Impact Factors, and Table 2 reports the number of 2013 citable items per journal.

Two independent curators divided the dataset and manually reviewed each journal's online author instructions and editorial policies between February 2016 and June 2016, and later spot checked each other's work. Because we were specifically interested in the information being communicated to manuscript submitting authors about data sharing requirements, we did not consider more peripheral sources of information, such as footnoted links to additional web pages, unless authors were specifically instructed to review this information in order to understand or comply with a journal's data sharing policy. We ranked the journals' data sharing policies using a rubric adapted from *Stodden, Guo & Ma (2013)* (Table 3). The Stodden Guo, and Ma rubric was utilized because it provided a starting scale for evaluating the current state of journal data sharing policies, as it was originally constructed to assess to what degree journals had implemented the National Academy of Sciences guidelines for data and code sharing. Moreover, the rubric was relatively easy to adapt and expand to assess policy characteristics specific to biomedical research, such as differentiating those policies that only addressed omics and structural data. Additionally, we examined the policies to determine the recommended data sharing method (e.g., a public repository or journal hosted), if data copyright or licensing recommendations were mentioned, the inclusion of instructions on how long the data should be made available, and if the policy noted reproducibility or analogous concepts. We chose to examine these characteristics based on their relationship to a growing body of best practices and recommendations associated with data sharing, such as those addressed in the TOP Guidelines (https://cos.io/our-services/top-guidelines/) that provision policy templates for journals. Finally, each journal was classified as either open access or subscription-based on its inclusion in the Directory of Open Access Journals database and this was confirmed on each journal's website (Table 4). If a discrepancy was found between these sources, we used the evaluation we gleaned from the journal's website. While we did not track the occurrence of the discrepancies, they were rare. Four independent curators were randomly assigned to evaluate the journal data sharing policies for 10 journals in our sample, 40 journals (12.5% of our sample) in total. The percentage of agreement and Cohen's kappa was calculated for each dimension of the rubric. Agreement between the original and independent curators ranged from 92.308% to 100%. The Cohen's kappa coefficient, which takes into account

**Table 1    Journal impact factor category.**

| Journal impact factor category | N (%) |
| --- | --- |
| <2 | 19 (6%) |
| 2–3.99 | 125 (39.3%) |
| 4–5.99 | 102 (32.1%) |
| 6–7.99 | 25 (7.9%) |
| 8–9.99 | 15 (4.7%) |
| 10–29.99 | 29 (9.1%) |
| 30–43 | 3 (0.9%) |

**Table 2    Number of citable items per journal.**

| Number of citable items per journal | N (%) |
| --- | --- |
| <100 | 42 (13.2%) |
| 100–500 | 239 (75.2%) |
| 500–1,000 | 28 (8.8%) |
| 1,000–32,000 | 9 (2.8%) |

the possibility of agreement occurring by chance and is generally considered a more robust measure, ranged from .629 to 1.0 (Table 5).

## Statistical methods

Continuous variables are summarized with medians and interquartile ranges (IQRs) denoting the 25th and 75th percentiles. Categorical variables are summarized with counts and percentages. The variables IF and total citable items are not normally distributed (Shapiro Wilk's Test $p$-values < 0.001), so medians are presented instead of means, and nonparametric methods are used for statistical tests.

The association of IF with 6-level data sharing mark (DSM) was tested with a nonparametric Kruskal-Wallis one-way analysis of variance (ANOVA) of IF in 2013 and 2014 with DSM as a grouping factor. Post-hoc pairwise two-sample Wilcoxon tests were used to determine whether the median IF for journals differ between the two-level data sharing policy (required vs. not required) categories. $P$-values from the Wilcoxon tests were adjusted for multiple comparisons with the Holm procedure.

Pearson's chi-square test was used to test the association of data sharing policy (two levels: required vs not required) and open access status. Fisher's Exact Test was used to test the association of the 6-level DSM with open access status. Fisher's Test was used as opposed to Chi-square test due to the low number of open access journals within some DSM categories. To examine the association of open access status and data sharing weighted by publishing volume we examined the number of citable items in each category and tested for the association of open access and data sharing with Pearson's chi-square test.

All statistical analyses were performed with R version 3.2.1 (*R Foundation for Statistical Computing RCT, 2016*). All code and data to reproduce these results can be found on GitHub (https://github.com/OHSU-Library/Biomedical_Journal_Data_Sharing_Policies).

| Table 3 | Journal scoring rubric used in this study, adapted from *Stodden, Guo & Ma (2013)*. |
|---|---|
| **Data sharing mark** | |
| 1 | Required as condition of publication, barring exceptions |
| 2 | Required but, no explicit statement regarding effect on publication/editorial decisions |
| 3 | Explicitly encouraged/addressed, but not required. |
| 4 | Mentioned indirectly |
| 5 | Only protein, proteomic, and/or genomic data sharing are addressed. |
| 6 | No mention |
| **Journal access mark (whole journal model, does not consider hybrid publishing)** | |
| 1 | Open access |
| 0 | Subscription |
| **Protein, proteomic, genomic data sharing required with deposit to specific data banks** | |
| a | Yes |
| b | No |
| **Recommended sharing method** | |
| A | Public online repository |
| B | Journal hosted |
| C | By reader request to authors |
| D | Multiple methods equally recommended |
| E | Unspecified |
| **If journal hosted** | |
| a | Journal will host regardless of size |
| b | Journal has data hosting file/s size limit |
| c | Unspecified |
| **Copyright/licensing of data** | |
| a | Explicitly stated or mentioned |
| b | No mention |
| **Archival/retention policy (statement about how long the data should be retained)** | |
| a | Explicitly stated |
| b | No mention |
| **Reproducibility or analogous concepts noted as purpose of data policy** | |
| a | Explicitly stated |
| b | No mention |

# RESULTS

Of the 318 journals examined, 38 (11.9%) required data sharing as a condition of publication and 29 (9.1%) required data sharing, but made no explicit statement regarding the effect on publication and editorial decisions. A total of 74 (23.3%) journals explicitly encouraged or addressed data sharing, but did not require it. 29 (9.1%) of journals mentioned data sharing indirectly. A total of 47 (14.8%) journals only addressed data sharing for proteomic, genomic data, or other specific omics data. A total of 101 (31.8% of journals did not mention anything about data sharing (Fig. 1 and Table 6).
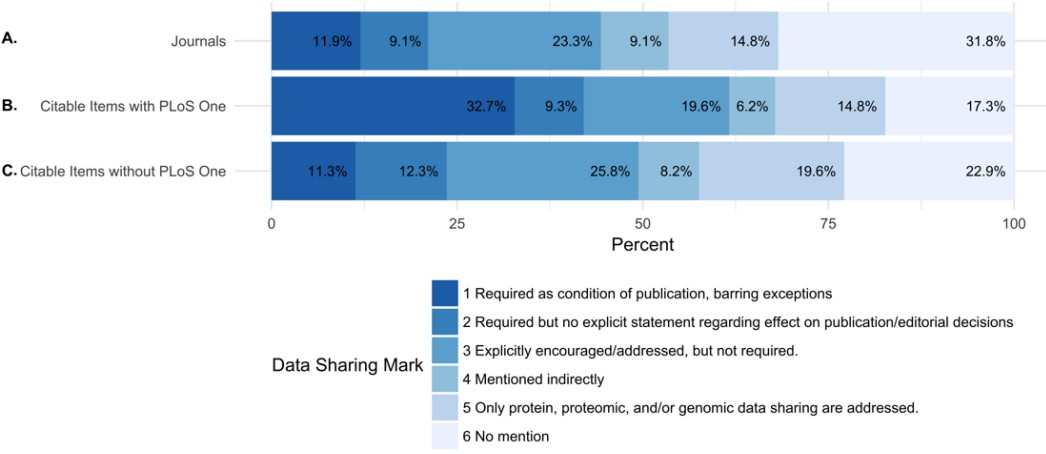

**Figure 1  Percentage of journals per each data sharing mark (DSM).** (A) shows the percentage of all journals for each data sharing mark. (B) shows the percentage of citable items from each journal (including PLOS ONE) for each data sharing mark. (C) shows the percentage of citable items for each journal (excluding PLOS ONE) for each data sharing mark. Because of the journal PLOS ONE's high publishing activity, we analyzed the percentage of citable items for each data sharing mark including and excluding PLOS ONE.

**Table 4  Number of journals per open access.**

| Open access | # Journals (%) | Median # citable items per journal 2013 | # Citable items 2013 (%) | # Citable items 2013, remove PLOS ONE (%) | Median # citable items per journal 2014 | # Citable items 2014 (%) | # Citable items 2014, remove PLOS ONE (%) |
|---|---|---|---|---|---|---|---|
| Open access | 44 (13.8%) | 199.5 | 43,789 (33.6%) | 12,293 (12.4%) | 207 | 45,831 (35.0%) | 15,791 (15.6%) |
| Subscription | 274 (86.2%) | 246.5 | 86,541 (66.4%) | 86,541 (87.6%) | 240 | 85,276 (65.0%) | 85,276 (84.4%) |

In order to understand the potential influence of the policies on the published literature, we also evaluated the distribution of publication volume by each data sharing mark. In 2013, the total number of citable items (papers) in the studied journals was 130,330. In 2014, the total number of citable items was 131,107. The median number of citable items per journal was 243.0 and 237.5, respectively (Table 5).

Table 5 shows the 2013 and 2014 publishing volume in citable items for each data sharing mark. While it is likely that some of the journals in the study implemented or revised their data sharing policies after 2014, the publishing volume data is current enough to provide an insight into the potential influence of existing journal data sharing policies on the published literature.

While only 21% of the journals in the study required data sharing (DSM 1 and 2), these journals published 42.1% of the citable items in 2013 and 2014 (23.6% and 24.9% of the citable items in 2013, 2014 after removing PLOS ONE) (Table 5).

The median 2013 journal IF for journals with the strongest data sharing policies (DSM 1) was 8.2; whereas, the median 2013 IF for journals with no mention of data sharing was 3.5. Figure 2 shows the median IF for each DSM category by report year. The IF wasalso analyzed by collapsing the DSM into two categories: Required (DSM 1, 2) and Not

**Table 5  Curator reliability.**

| Score | # Journals | % Agreement | Cohen's Kappa |
| --- | --- | --- | --- |
| Open access mark | 40 | 100.000 | 1.000 |
| Data sharing mark | 40 | 92.500 | 0.905 |
| Protein proteomic genomic or microaray sequence or structural data sharing addressed required with deposit to specific data banks | 40 | 100.000 | 1.000 |
| Recommended preferred sharing mark | 40 | 97.500 | 0.959 |
| Size guidelines if journal hosted provided | 13 | 92.308 | 0.629 |
| Copyright licensing mark | 40 | 100.000 | 1.000 |
| Archival retention mark | 40 | 97.500 | 0.655 |
| Reproducibility noted mark | 40 | 92.500 | 0.754 |

Required (DSM 3, 4, 5, 6). The median 2013 IF for the journals that required data sharing was 6.8, and the median 2013 IF for the journals that did not require data sharing was 4.0.

Impact Factor is significantly associated with the six-category data sharing marks (Kruskal-Wallis rank sum test, 5 $df$, $p < 0.001$, 2013 and 2014). Examining pairwise differences between DSM categories, we see that journals with DSM 1 have significantly higher IF than journals with DSM 3, 4, 5, or 6 (Wilcoxon test, $p < 0.001$, <0.001, 0.04, <0.001; 2013 data, 2014 similar). Journals with DSM 2 have significantly higher IF than journals with DSM 3, 4, or 6 (Wilcoxon test, $p = 0.034$, 0.0072, 0.0033; 2013 data, 2014 similar). Journals with DSM 5 have significantly higher IF than journals with DSM 3, 4, and 6 (Wilcoxon test, $p$ 0.0022, <0.001, <0.001; 2013 data, 2014 similar). In general, IF is not significantly different between DSM 1 and 2 and between DSM 2 and 5, reflecting the similar IF for journals with explicit data sharing requirements, either full or partial sharing. After collapsing DSM into two categories, required (DSM 1, 2) and not required (DSM 3, 4, 5, 6), we still see a highly significant increase in IF for journals with required data sharing (Wilcoxon Rank Sum Test, $p < 0.001$, 2013 and 2014 data) (Fig. 2).

Table 7 shows the count of subscription and open access journals for each DSM category, and the count and percentage of subscription and open access journals for each DSM category. The Fisher's Exact Test result, which yielded a $p$-value of 0.07, showed no significant association between the DSM and a journal's access model. We also tested this association by collapsing the DSM into two categories, required (DSM 1, 2) and not required (DSM 3, 4, 5, 6), and using a Chi-square test. Again, no significant association was found (Chi-square Test, $df = 1$, $p = 0.62$). Both results suggest that journals with a data sharing requirement are not more likely to be open access than journals without a data sharing requirement.

Although there was no significant association between open access and DSM at the journal level, we observed a highly significant association at the citable item level (Chi-square Test, $df = 1$, $p < 2e - 16$). That is, a citable item that is open access is much more likely to be published in a journal with a data sharing requirement (DSM 1 or 2). The proportion of open access journals that require data sharing is much larger than the proportion of subscription journals (64.3% vs 11.3%). The very small $p$-value is partially

Vasilevsky et al. (2017), *PeerJ*, DOI 10.7717/peerj.3208

**Table 6 Publishing volume by data sharing mark.**

| DSM | DSM description | # Journals (%) | Median # citable items per journal 2013 | # Citable items 2013 (%) | # Citable items 2013, remove PLOS ONE (%) | Median # citable items per journal 2014 | # Citable items 2014 (%) | # Citable items 2014, remove PLOS ONE (%) |
|---|---|---|---|---|---|---|---|---|
| 1 | Required as condition of publication, barring exceptions | 38 (11.9%) | 230.5 | 42,669 (32.7%) | 11,173 (11.3%) | 220 | 42,794 (32.6%) | 12,754 (12.6%) |
| 2 | Required but no explicit statement regarding effect on publication/editorial decisions | 29 (9.1%) | 209 | 12,138 (9.3%) | 12,138 (12.3%) | 227 | 12,436 (9.5%) | 12,436 (12.3%) |
| 3 | Explicitly encouraged/addressed, but not required. | 74 (23.3%) | 259.5 | 25,519 (19.6%) | 25,519 (25.8%) | 282.5 | 26,026 (19.9%) | 26,026 (25.8%) |
| 4 | Mentioned indirectly | 29 (9.1%) | 256 | 8,062 (6.2%) | 8,062 (8.2%) | 225 | 7,894 (6%) | 7,894 (7.8%) |
| 5 | Only protein, proteomic, and/or genomic data sharing are addressed. | 47 (14.8%) | 277 | 19,339 (14.8%) | 19,339 (19.6%) | 316 | 19,080 (14.6%) | 19,080 (18.9%) |
| 6 | No mention | 101 (31.8%) | 211 | 22,603 (17.3%) | 22,603 (22.9%) | 213 | 22,877 (17.4%) | 22,877 (22.6%) |
| *Publishing volume by data sharing requirement* | | | | | | | | |
| DSM 1&2 | Required | 67 (21.1%) | 226 | 54,807 (42.1%) | 23,311 (23.6%) | 221 | 55,230 (42.1%) | 25,190 (24.9%) |
| DSM 3–6 | Not Required | 251 (78.9%) | 248 | 75,523 (57.9%) | 75,523 (76.4%) | 244 | 75,877 (57.9%) | 75,877 (75.1%) |
| *Publishing volume in all journals* | | | | | | | | |
| **Total** | All Journals | 318 (100%) | 243 | 130,330 (100%) | 98,834 (100%) | 237.5 | 131,107 (100%) | 101,067 (100%) |
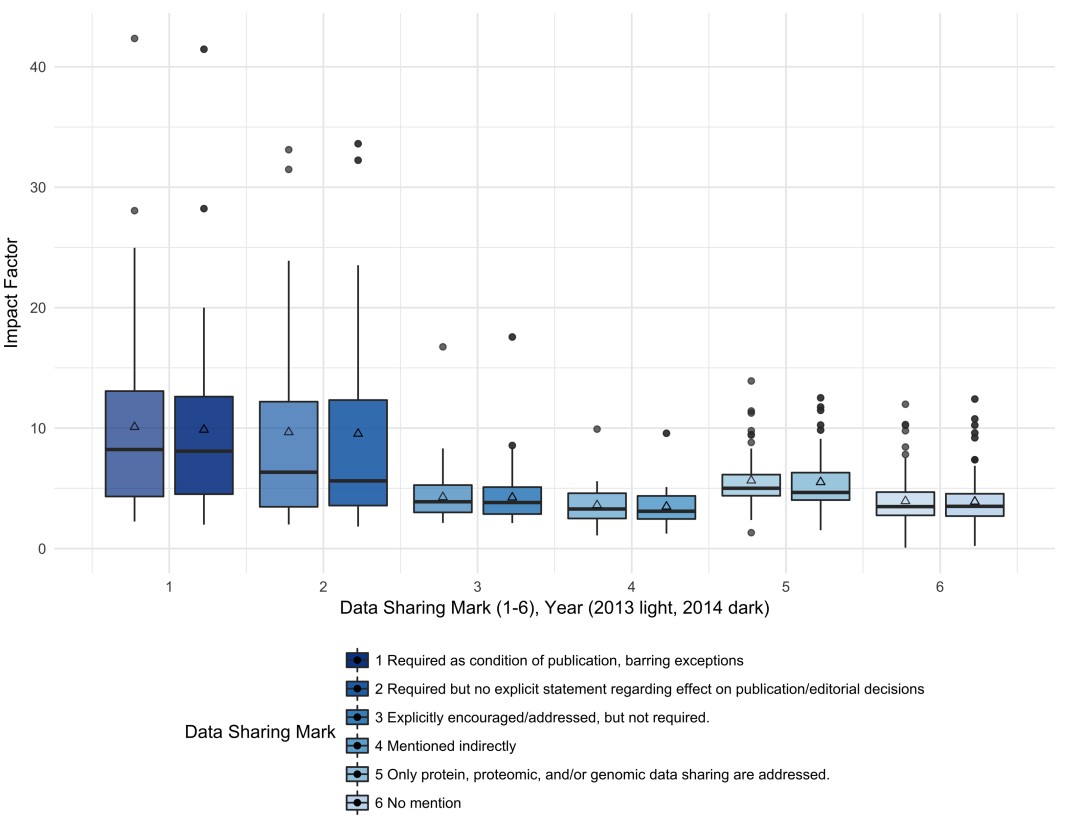

**Figure 2 Impact factors were higher for journals with the strongest data sharing policies (DSM 1) compared to journals with no mention of data sharing (DSM 6).** The median Impact Factor was calculated for the journals with each data sharing mark (1–6) for each report year (left = 2013, right = 2014). The lower and upper hinges of the boxplots represent the first and third quartiles of journal Impact Factor, the horizontal line represents the median, the triangle represents the mean, and the upper and lower whiskers extend from the hinge to the highest (lowest) value that is within 1.5 times the interquartile range of the hinge, with journals outside this range represented as points.

due to the large number of total citable articles studied and also due to the large proportion of open access citable items in PLOS ONE. However, even with PLOS ONE removed from the analysis, an open access article is still more likely to have been published in a journal with a data sharing requirement and the proportion of open access journals versus subscription journals that require data sharing is 16.0% vs 11.3% (Chi-square Test, $df = 1$, $p < 2e - 16$).

As illustrated in Fig. 3, excluding those journals with no mention of data sharing (DSM 6), 57.6% (125) of the journals in the data set recommended data sharing via a public repository, 20.7% (45) recommended sharing via a journal hosted method, 1.8% (4) recommend sharing by reader request to authors, 5.1% (11) state multiple equally recommended methods and 14.8% (32) do not specify.

Of the journals requiring data sharing (DSM 1 or 2), 85% (57) recommend data sharing via a public repository. Of the journals that recommended data sharing via a journal hosted method, the majority, 88.8% (40), did not specify any size limitations.

**Table 7  Open access journals and citable items by data sharing mark.**

| Open access journals and citable items by data sharing mark | Subscription | Open access | % Open access |
|---|---|---|---|
| Data sharing mark | # Journals (# Citable items) | # Journals (# Citable items) | % Journals (% Citable items) |
| 1- Required as condition of publication, barring exceptions | 29 (7,709) | 9 (34,960; 3464[a]) | 23.7% (81.9%; 31%[a]) |
| 2- Required but no explicit statement regarding effect on publication/editorial decisions | 27 (11,864) | 2 (274[a]) | 6.9% (2.3%) |
| 3- Explicitly encouraged/addressed, but not required. | 63 (22,884) | 11 (2,635) | 14.9% (10.3%) |
| 4- Mentioned indirectly | 29 (8,062) | 0 (0) | 0% (0%) |
| 5- Only protein, proteomic, and/or genomic data sharing are addressed. | 40 (17,401) | 7 (1,938) | 14.9% (10.0%) |
| 6- No mention | 86 (18,621) | 15 (3,982) | 14.9% (17.6%) |
| *Data sharing requirement* | | | |
| DSM 1&2 - Required | 56 (19,573) | 11 (35,234; 3,738[a]) | 16.42% (64.29%; 16.04%[a]) |
| DSM 3–6 - Not required | 218 (66,968) | 33 (8,555) | 13.15% (11.33%) |

**Notes.**
[a]After removing PLOS ONE.

Only 7.3% (16) journals that addressed data sharing (DSM 1,2,3, 4, and 5) explicitly mentioned copyright or licensing considerations. Even for those journals that required data sharing (DSM 1 or 2), only 16.4% (11) mentioned copyright or licensing; however, these journals published 31.9% of the citable items in 2013 of the journals that addressed data sharing. Only 2 journals in the entire data set addressed how long the data should be retained.

In light of its frequently used justification, we also coded the data sharing policies for a mention of scientific reproducibility or analogous concepts. Reproducibility or similar language was mentioned by 16.9% (54) of the total studied journals. Of the journals requiring data sharing (DSM 1 or 2), 65.5% (44) mentioned the concept of reproducibility.

## DISCUSSION

Publishers have an influential role to play in promoting, facilitating, and enforcing data sharing (*Dallmeier-Tiessen et al., 2014*; *Lin & Strasser, 2014*). However, only a minority of the journals analyzed for this this study required data sharing. While the capacity of the existing policies is more promising if considered from the perspective of publishing volume, our results were consistent with other examinations of data sharing policies (*Piwowar, Chapman & Wendy, 2008*; *Barbui, 2016*; *McCain, 1995*). Like Piwowar and Chapman (*Piwowar, Chapman & Wendy, 2008*), we found that a large proportion of the journals we examined (40%) required the deposition of omics data to specific repositories. Less frequent and more varied, however, were requirements that addressed data in general. The higher prevalence of omics data sharing requirements we observed may be due to the more mature guidelines, reporting standards, and centralized repositories for omics data types (*Piwowar, Chapman & Wendy, 2008*; *Brazma et al., 2001*; *Hrynaszkiewicz et al., 2016*; *Piwowar & Chapman, 2010*). The further development and implementation of well communicated best practices and resources for general data types, could be a means for

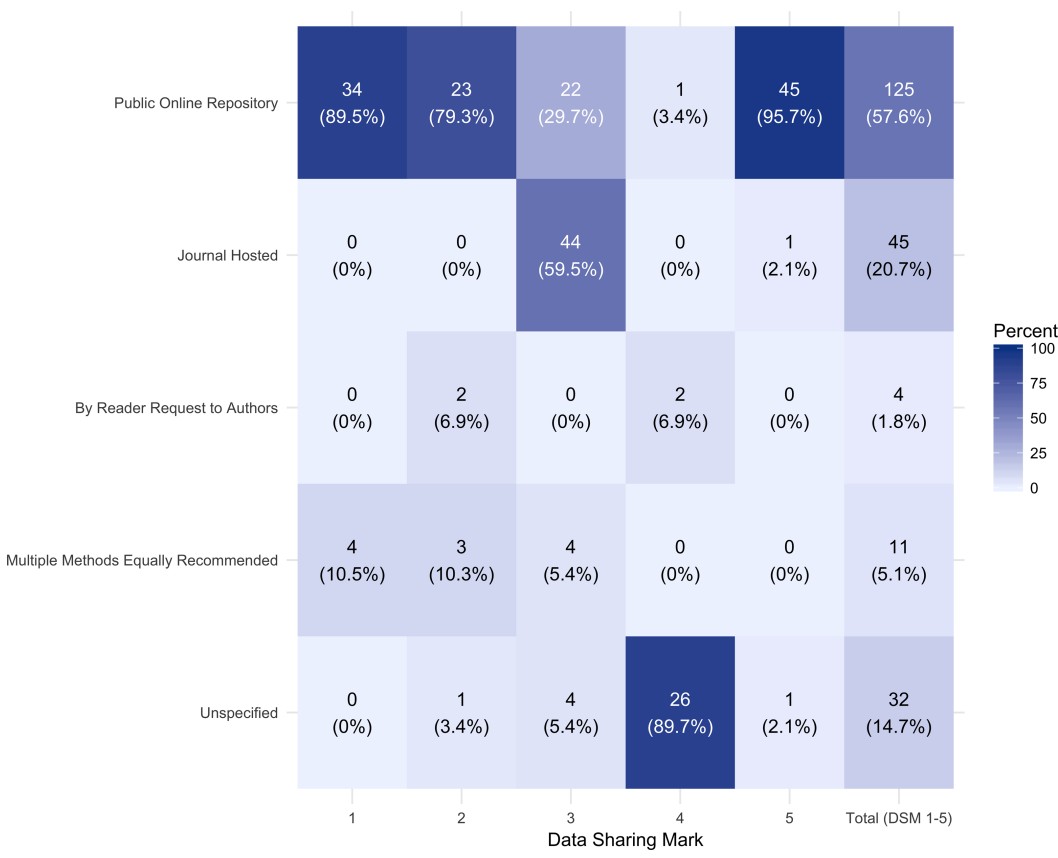

**Figure 3** **Recommended data sharing method by data sharing mark (DSM) 1–5.** The number (percent) of journals with each recommended data sharing method is represented by each tile, with brighter blue shades denoting higher percentages of journals with the given data sharing method.

increasing the prevalence and strength of journal data sharing requirements and ensuring compliance (*Lin & Strasser, 2014*).

While a problematic and often abused proxy for quality, the IF is closely associated with a journal's prestige (*Lariviere et al., 2016*). It influences publication decisions and the perceived significance of individual papers (*Lariviere et al., 2016*). Because of its impact on scholarly communication, it is noteworthy that there was a significantly higher IF associated with the journals with a data sharing requirement. This result was similar to other studies (*Piwowar, Chapman & Wendy, 2008*; *Stodden, Guo & Ma, 2012*; *Sturges et al., 2015*; *Magee, May & Moore, 2014*). As has been noted, prestigious journals may be better positioned and be more willing to impose new requirements and practices on authors (*Lin & Strasser, 2014*; *Piwowar, Chapman & Wendy, 2008*).

The importance and benefits of data sharing are often linked to and discussed within the larger context of open access to research results, specifically the published literature. Public access to both peer reviewed articles and data are regarded as necessary elements for addressing problems within the scientific enterprise and realizing the full value of research investments, as exemplified in recent policies aimed at increasing access to and the quality of biomedical research in which both open access and data sharing are addressed

(*Holdren, 2012*). As such, we wanted to investigate if there was a positive relationship between open access journals and data sharing requirements. While we found that an open access citable item is much more likely to be published in a journal with a data sharing requirement, we did not find that open access journals are any more likely to require data sharing than subscription journals. This result is in contrast with a previous finding from Piwowar and Chapman (*Piwowar, Chapman & Wendy, 2008*). However, we analyzed a greater number of journals and a greater number of open access journals. We hypothesize some open access journals may be less willing to impose additional requirements, because they lack the prestige or prominence of more established journals and publishers. Smaller and independent open access journals may also lack the resources to facilitate and enforce data sharing

How data is managed and shared affects its value. If a data set is difficult to retrieve or understand, for example, replication studies can't be performed and researchers can't use the data to investigate new questions. While 65.7% of the journals in our study that required data sharing addressed the concept of reproducibility, as with earlier investigations (*Piwowar, Chapman & Wendy, 2008*; *Sturges et al., 2015*) we found that most data sharing policies did not provide specific guidance on the practices that ensure data is maximally available and reusable (*DataONE, 2016*; *Starr et al., 2015*). For example, the majority of journals that addressed data sharing (DSM 1–5) recommended depositing data in a public repository; however, only a handful of journals provided guidelines or requirements related to licensing considerations or retention timeframes. While a higher IF was associated with the presence of a data sharing requirement, overall the policies did not provide guidelines or specificity to facilitate reproducible and reusable research. This result is similar to a previous study in which we showed that the majority of biomedical research resources are not uniquely identifiable in the biomedical literature, regardless of journal Impact Factor (*Vasilevsky et al., 2013*).

Our study confirms earlier investigations which observed that only a minority of biomedical journals require data sharing, and a significant association between higher Impact Factors and journals with a data sharing requirement. Our approach, however, included several limitations. Only journals in the top quartiles by volume or Impact Factor for the Web of Science categories we identified as belonging to the biomedical corpus were analyzed, which introduced some inherent biases. While the 2014 median Impact Factor for the journals in our sample was 4.16, the median Impact Factor for all journals in the Web of Science categories we utilized was 2.50. Similarly, the median number of 2014 citable items was 237.5 for our sample, and 94.0 for all journals. In hindsight, it would have been valuable to have systematically analyzed more nuanced aspects of the policies' quality characteristics, such as whether minimal information or metadata standards were addressed and if the shared data was reviewed in the peer review process. Finally, it should be noted that many of the policies we reviewed were difficult to interpret. While the study's authors are confident that the data sharing scores we assigned reflect the most accurate interpretation of each journal's policy at the time of our data collection, the policies in general included ambiguous and fragmented information. It is possible, therefore, that there are gaps between the scores we assigned to some policies and their editorial intent.

There have also been changes to some policies since our last data collection. Springer, for example, implemented a new policy in July of 2016 as described by *Freedman, 2016* (http://www.springersource.com/simplifying-research-data-policy-across-journals/).

As a continuation of this work, we plan to maintain a community curated and regularly updated public repository of journal data sharing policies, which will include journals beyond the sample represented in this study. The repository will utilize a schema that builds upon the rubric used for this study, but also addresses additional and more nuanced aspects of data sharing, such as those described above and addressed in the TOP Guidelines (*The TOP Guidelines Committe, 2016*), the FAIR Data Principles (*Wilkinson et al., 2016*), as well as the FAIR-TLC metrics, which build upon the FAIR Data Principles to also make data Traceable, Licensed, and Connected (*Haendel et al., 2016*). We will also collect additional information about the journals, such as age and frequency of publication, as outlined by *Stodden, Guo & Ma (2012)*. In addition to providing a queryable resource of journal data sharing policies, the database's curation schedule will facilitate an understanding of policy changes over time and inform future evolution. It is our hope that data from the repository will be ingested and used by other resources, such as BioSharing (*McQuilton et al., 2016*), to further the discovery and analysis of data sharing policies. A follow-up study will look at the data availability for articles associated with the journals in this study. Finally, building upon recommendations outlined by the Journal Research Data (JoRD) Project (*Sturges et al., 2015*), Lin and Strasser (*Lin & Strasser, 2014*), the Center for Open Science (https://cos.io/), and the FAIR-TLC metrics, we intend to convene a community of stakeholders to further work on recommendations and template language for strengthening and communicating journal data sharing policies. Maximally available and reusable data will not be achieved via the implementation of vague data sharing policies that lack specific direction on where data should be shared, how it should be licensed, or the ways in which it should be described. On the contrary, such specificity is essential.

## CONCLUSIONS

We observed a two-pronged problem with journal data sharing policies. First, given the attention the benefits of data sharing have received from the biomedical community, it is problematic that only a minority of journals have implemented a strong data sharing requirement. Second, among the policies that do exist, guidelines vary and are relatively ambiguous. Overall, the biomedical literature is lacking policies that would ensure that underlying data is maximally available and reusable.

This is problematic if we are to realize the outcomes and improvements that open data is supposed to facilitate.

## ACKNOWLEDGEMENTS

Thanks to the following colleagues for their curation assistance or visualization advice: Steven Bedrick, PhD; Heather Coates, MLIS; Jill Emery, MLIS; Erin Foster, MLIS; Danielle Robinson; Chris Shaffer, MLIS; Kate Thornhill, MLIS; Jackie Wirz, PhD.

### Funding

This work was supported by National Institutes of Health grant numbers NIH BD2K SCC HHSN316201200001W, HHSN27200001. The funders had no role in study design, data collection and analysis, decision to publish, or preparation of the manuscript.

### Grant Disclosures

The following grant information was disclosed by the authors:
National Institutes of Health: NIH BD2K SCC HHSN316201200001W, HHSN27200001.

### Competing Interests

The authors declare there are no competing interests.

### Author Contributions

- Nicole A. Vasilevsky and Robin E. Champieux conceived and designed the experiments, performed the experiments, wrote the paper, reviewed drafts of the paper.
- Jessica Minnier analyzed the data, contributed reagents/materials/analysis tools, prepared figures and/or tables, reviewed drafts of the paper.
- Melissa A. Haendel reviewed drafts of the paper.

### Data Availability

  Github: https://github.com/OHSU-Ontology-Development-Group/Biomedical_Journal_Data_Sharing_Policies.

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
