# Peer review of "Reproducible and reusable research: are journal data sharing policies meeting the mark?"

_PeerJ, doi:10.7717/peerj.3208_

## Round 0.1 · original submission · Major Revisions

Please pay special attention to the comments of the Reviewer #3.

·

Basic reporting

The article is well written and uses appropriate terminology and good English. There is a good introduction which sets the context well and explains both the importance of data sharing and the role journals can play. The figures are relevant, clear and well presented.

The literature has been quoted well and from a reasonable breadth of sources. However I was surprised to see no mention of the Center for Open Science and the TOP Guidelines

The raw data is provided at Github with link in the article. The data elements have been well organised and are easy to access.

Experimental design

The authors describe a piece of original primary research which addresses an important issue which they have framed precisely - to investigate the prevalence and characteristics of data sharing policies in the biomedical research literature.

The methods are well described and in sufficient detail to allow other workers to replicate the study.

The rubric used to assess the strength of journals’ data sharing policies was a modification of that of Stodden, Guo and Ma (2013). However, no justification for this was offered. There exists already a good protocol in the form of the TOP Guidelines from the Center for Open Science (https://cos.io/top/#summary) which could have been used. I was surprised to see no mention of this, especially as it is an ‘off the shelf’ rubric.

Journals were classified as open access or not based on the DOAJ. However, the accuracy of this source has been called into question a number of times (e.g. Nature News, 6 August 2014). No reason was given for using DOAJ rather than simply making a determination from the websites of the journals (which were being consulted in any case to find the data policies).

The authors restricted the selection of journals to those in the top quartile either by published articles or impact factor, yet they do not state why they made this cut. Perhaps it was on the basis that these would be the ‘most important’ journals, or perhaps to keep the task to a manageable size (given the manual review of the journals). The limits of this selection are raised in the discussion.

It is not stated whether the manual review of the journals’ policies was conducted by a single investigator or by several. If the latter, how was any operator effect controlled for? This is important given the lack of clarity of many publishers policies as different investigators may have made different assessments.

There are no ethical issues of relevance.

Validity of the findings

I am not qualified to comment on the statistical procedures used so it would be important to ensure a statistical review also.

The findings appear to be supported by the data which has been analysed in an appropriate way given the research question. The authors have provided a good answer to the question they set themselves, with the proviso that they were looking at a selection of the biomedical literature only.

It is a pity that they didn’t analyse the whole corpus of 1166 journals in their initial selection as the picture may have been different. The un-analysed journals (representing the remaining 39% percent of biomedical articles) would have given a fuller picture.

The authors highlight the important point that publishers are not explaining their data policies well. Their stated plans for further research cover some important topics.

Additional comments

Line 60: “on” should read “of”

Line 108: “World” should read “Web”

Line 211-213: This sentence is a tautology since the two statements in this sentence mean the same thing. Yet the sentence gives the impression that they are independent

Line 297: “World” should read “Web”

·

Basic reporting

Abstract: in Results what happened to the rest of 23.9% of journals?
Introduction Ln 92: there is an extra quotation mark at the end of the sentence.
Results: Lns 167-171: What happened to the rest of 130 (41%) journals?

Experimental design

Materials and Methods: Ln 126-127 "We manually reviewed each journal’s online author instructions and editorial policies between February 2016 and June 2016." It is not clear how many authors checked the sites, and if the sites were checked only once.
Lns 302-303 – “Finally, it should be noted that many of the policies we reviewed were difficult to interpret.” For such issues it would be plausible to have at least 20-30% data extractions in pairs, and then have inter-rater agreement checked.
These are important limitations of the study and should be addressed.

Validity of the findings

No comments.

Additional comments

Consider shortening.

Reviewer 3 ·

Basic reporting

The authors have satisfied the basic reporting standards of PeerJ.

Experimental design

Authors say that “the study was designed to determine the pervasiveness and quality of data sharing policies.” The authors were able to achieve the former (pervasiveness) of policy, but no attention was paid to the quality of the policy per se (i.e., the latter). They have combed through online information and captured the presence of a data sharing policy where the extent of coverage across certain dimensions is noted. But policy quality involves a review of how well the policy performed and/or how accurate. Quality refers to accuracy, appearance, usefulness, or effectiveness. To analyze it, authors would need to look at the policies’ effectiveness, unintended effects, equity (effects across different populations within sub-disciplines of biomed or across countries). Or they may examine implementation costs, implementation feasibility, and acceptability by authors/community receptiveness. If this expands the scope of the paper beyond the author's intent or interest, authors could drop or explicitly qualifying the term "quality."

Open Access & versus data sharing:
There are rhetorical connections between the two which have been made in certain environments, but there are no necessary connections. It is true that OA has been linked to data sharing as an advocacy tactic within the OA community, but it doesn't follow that there is an intrinsic association, especially since data sharing supports the scientific process of validating and vetting knowledge claims that applies equally across biomed research (and research in general). I would like to see more context in the paper then as to why it's useful to examine data sharing policies between OA and nonOA journals. (Authors refer to the Piwowar study on articles published in 2007, but given the progress made in data sharing in the past few years, perhaps this is not sufficient enough as an explanation.)

Validity of the findings

* I have not conducted statistical review of the findings nor attempted to reproduce the results.

* If the author's aims are to understand the potential influence of the policies on the published literature, then the timing of the data collected and analyzed is not lining up for me. Authors reviewed policies in 2015 and 2016, but they are analyzed with 2013 and 2014 publication counts and 2013 JIF. The time difference is even a bigger issue considering the considerable changes that have occurred in the past few years thanks to the growing awareness of data sharing, availability of resources to follow suit, etc. The manuscript does not specify when these policies were put into place. Many of the policies reviewed in 2015/2016 may have changed since 2013, if not wholesale (new policy) or in part (clearer instructions).

* There is reasonable amount of evidence out there that JIF is a reasonable proxy for journal prestige as determined by surveys of experts, and so using it to ask how the prestige of a journal is associated with quality of process is not completely off mark. I appreciate that the authors prefaced the JIF discussion with a note acknowledging the shortcomings of JIF as a proxy for research article quality. But the biggest problem is that all these different things are lagging indicators to differing degrees so its impossible to disentangle whether your just getting a signal of change over time, which is almost always a massive confounder for any of these studies. There are better ways to ask the question and better (if more resource intensive) ways of identifying how a community views a set of journals.

The Stodden paper conclusion contains an outline of limitations and future areas to explore, which are very instructive for this study: "This study models open data and code policy adoption, using impact factor and publishing house as explanatory variables, but research could be carried out using a more extensive set of confounding variables such as field characteristics, journal size, journal age, frequency of publication, proportion of computational results published in the journal, proportion of computational results publishing in the field." As these have been noted already, I encourage authors to explore these additional routes as a way to build on previous research. Authors could then close the time gap issue by updating the analysis to the point in which these editorial policies were reviewed.

Additional comments

* What does it mean to require data sharing - need an explicit definition wherever term is used (or once, upfront). Is it a binary or is it a spectrum of levels? For multi-level model, see SpringerNature: http://blogs.nature.com/ofschemesandmemes/2016/07/05/promoting-research-data-sharing-at-springer-nature and http://www.springernature.com/gp/group/data-policy/.

* There are already databases that contain community crowdsourced and curated data sharing policies (ex: BioSharing https://biosharing.org/). If the authors consider these insufficient, whereby their own is worth creating, please indicate as much (and why).

* Please properly cite code/data per FORCE11 Data Citation Principles, Software Citation Principles. While they are available in Github, I strongly encourage authors to follow researcher best practices to preserve and make software citeable in a sustainable, identifiable and simple way. For example: the Github - Zenodo integration makes this incredibly easy to do.

---

## Round 0.2 · Minor Revisions

Thank you for taking into consideration the reviewers' comments and suggestions. There are a few remaining comments which should be addressed.

·

Basic reporting

see below

Experimental design

see below

Validity of the findings

see below

Additional comments

I am happy that the authors have adequately addressed the issues I raised in my initial review. They have answered my queries satisfactorily and have added extra text in the article for the purposes of clarification/explanation. I have no further concerns and recommend publication.

·

Basic reporting

Pass.

Experimental design

LN 156 “Finally, each journal was classified as either open access or subscription-based on its inclusion in the Directory of Open Access Journals database and this was confirmed on each journal’s website.”
Please clarify: how were the journals classified if the information on journal’s website did not match DOAJ classification and what were the numbers of journals matching and journals not matching.

Validity of the findings

Table 6, column “#Citable Items 2013, Remove PLoS One (%)” should be revised (raw “4” seems that comma is misplaced).
Table 7, column “Open Access" should be revised (it seems asterisk is missing by 274).

Reviewer 3 ·

Basic reporting

The article satisfies PeerJ's basic reporting standards.

Experimental design

The authors have taken account my concerns about the experimental design in the latest version.

Validity of the findings

The authors have taken account my concerns about the validity of the findings in the latest version.

Additional comments

I am happy to recommend publication for this manuscript on the condition that the authors comply with the PeerJ data and materials sharing policy.
"PeerJ is committed to improving scholarly communications and as part of this commitment, all authors are responsible for making materials, code, data and associated protocols available to readers without delay. The preferred way to meet this requirement is to publicly deposit as noted below. Cases of non-compliance will be investigated by PeerJ, which reserves the right to act on the results of the investigation."

I was not satisfied with their answer to deposit after publication. The authors say in their rebuttal: "We will archive all the code and data in Zenodo after publication." This does not satisfy the data policy, and so I would like to see that data is deposited BEFORE publication with the data cited properly in the paper. (Policy: "In all cases, accession / deposition reference numbers must be provided in the manuscript.")

---

## Round 0.3 · accepted · Accept

Thank you for addressing the final comments of the reviewers.